# Enhanced β-carotene and Biomass Production by Induced Mixotrophy in *Dunaliella salina* across a Combined Strategy of Glycerol, Salinity, and Light

**DOI:** 10.3390/metabo11120866

**Published:** 2021-12-13

**Authors:** Willian Capa-Robles, Ernesto García-Mendoza, José de Jesús Paniagua-Michel

**Affiliations:** 1Department of Marine Biotechnology, Centro de Investigación Científica y de Educación Superior de Ensenada (CICESE), 22860 Ensenada, Baja California, Mexico; wcapa@cicese.edu.mx; 2Department of Biological Oceanography, Centro de Investigación Científica y de Educación Superior de Ensenada (CICESE), 22860 Ensenada, Baja California, Mexico; ergarcia@cicese.mx

**Keywords:** *Dunaliella*, glycerol, β-carotene, photosynthesis, mixotrophy

## Abstract

Current mixotrophic culture systems for *Dunaliella salina* have technical limitations to achieve high growth and productivity. The purpose of this study was to optimize the mixotrophic conditions imposed by glycerol, light, and salinity that lead to the highest biomass and β-carotene yields in *D*. *salina*. The combination of 12.5 mM glycerol, 3.0 M salinity, and 50 μmol photons m^−2^ s^−1^ light intensity enabled significant assimilation of glycerol by *D*. *salina* and consequently enhanced growth (2.1 × 10^6^ cell mL^−1^) and β-carotene accumulation (4.43 pg cell^−1^). The saline and light shock induced the assimilation of glycerol by this microalga. At last stage of growth, the increase in light intensity (300 μmol photons m^−2^ s^−1^) caused the β-carotene to reach values higher than 30 pg cell^−1^ and tripled the β-carotene values obtained from photoautotrophic cultures using the same light intensity. Increasing the salt concentration from 1.5 to 3.0 M NaCl (non-isosmotic salinity) produced higher growth and microalgal β-carotene than the isosmotic salinity 3.0 M NaCl. The mixotrophic strategy developed in this work is evidenced in the metabolic capability of *D. salina* to use both photosynthesis and organic carbon, viz., glycerol that leads to higher biomass and β-carotene productivity than that of an either phototrophic or heterotrophic process alone. The findings provide insights into the key role of exogenous glycerol with a strategic combination of salinity and light, which evidenced unknown roles of this polyol other than that in osmoregulation, mainly on the growth, pigment accumulation, and carotenogenesis of *D. salina*.

## 1. Introduction

Microalgae are one of the primary aquatic resources with the potential for exploitation and cultivation to obtain economically valuable compounds. The chlorophyte *Dunaliella salina* is the supreme and natural source of β-carotene, a carotenoid with bioactive roles such as provitamin *A*, colorant, antioxidant, antimicrobial, pro-apoptotic, anti-aging immunomodulator, and anticarcinogenic [1,2]. The biomass and β-carotene of this microalga are used extensively in the food, feed, pharmaceutical, biomedical, and nutraceutical and cosmeceutical industries [3,4]. Moreover, this halophilic microorganism is a promising source of other carotenoids (zeaxanthin, lutein, and α-carotene), vitamins, lipids, and proteins [3,5,6]. For a long time, the uniqueness of *D*. *salina* to thrive under extreme environmental conditions has been worldwide exploited at photoautotrophic production systems. However, yield and productivity under this practice are low due to photoinhibition, photo-oxidative damage, and self-shading of cultures intended to produce high cell densities [7]. Mixotrophic systems are attractive due to their low light requirements and large availability of carbon (inorganic and organic), contributing to improved growth rate, biomass, and extensive synthesis of metabolites. Mixotrophy involves the concomitant, simultaneous, and efficient metabolism between photosynthesis and respiration. Consequently, various metabolic pathways and sub-routes to incorporate and allocate carbon and energy are activated in microalgae [8,9]. Today, organic carbon nutrition and mixotrophy linked to metabolic pathways represent new scenarios in expanding knowledge and diversification of the metabolism and physiology of microalgae of environmental and economic importance.

Efforts to develop mixotrophic growth media for *D. salina* and many other microalgae are currently underway. The growth of *D. salina* on glucose [10,11,12] and acetate [10,13,14] supplemented media was reported, but they still exhibit predominantly low growth rates or a very low amount of β-carotene. Regarding substrates, glucose has the disadvantage of reducing photochemical O_2_ evolution, cellular affinity for CO_2_, and the synthesis of the Calvin cycle enzymes in microalgae [15]. This sugar is relatively expensive compared to other organic substrates [16]. Acetate can be toxic in high concentrations for various microorganisms [17] and sometimes is an inadequate substrate for β-carotene synthesis in microalgae [13]. In recent years, the use of glycerol for the mixotrophic production of biomass and carotenoids from microalgae, including *D*. *salina*, has been an important initiative by some groups of research [18,19,20,21]. Glycerol can be assimilated properly under respiratory conditions by photosynthetic microorganisms [22]. Moreover, this polyol lacks toxic effects on microalgal cells even at high concentrations [23]. This C_3_ molecule is the major by-product of the biodiesel industry (known as crude glycerol). This hypersaline waste could promote the growth and production of halophilic carotenogenic microorganisms [24]. Meanwhile, *D*. *salina* synthesizes glycerol to succeed in media and environments containing NaCl concentrations from about 0.05 M to saturation (around 5.5 M) [25,26]. Unlike other organic carbon substrates, the role of glycerol on the physiology and metabolism of *D*. *salina* may involve increased cell growth, β-carotene synthesis, and carotenogenesis.

Currently, entrepreneurs and companies have difficulties in implementing mixotrophic microalgal production under optimal culture conditions to maximize the yield and productivity in this biosystem [27]. Therefore, the substrate concentration and the environmental factors that best stimulate the mixotrophic metabolism of *D*. *salina* on glycerol must be optimized. Likewise, the role of glycerol on generating more biomass, β-carotene synthesis and carotenogenesis from *D*. *salina* must be explained. To investigate these gaps in knowledge of the induced mixotrophic metabolism, we assessed how glycerol, salinity, and light intensity under optimized conditions can stimulate the simultaneous and competitive production of biomass and β-carotene in this microalga under mixotrophic culture conditions. This paper presents the individual and combined effect of glycerol level, salinity, and light intensity on the nutritional and metabolic capacities of *D*. *salina* to maximize the cellular growth and carotenoids content under mixotrophy. Moreover, this research reveals for the first time the relationship between growth and carotenogenesis, especially of β-carotene in *D*. *salina* cultured mixotrophically on glycerol.

## 2. Results

### 2.1. Effect of Glycerol Concentration on Mixotrophic Culture of D. salina under 1.5 M NaCl Isosmotic Salinity

Due to the heterogeneity of both the salinities and glycerol levels, and the cell growth reported from previous studies [14,20,28], the effect of glycerol concentration on the culture of *D. salina* under better growth isosmotic conditions (1.5 M NaCl) [29] was evaluated. This saline condition and 12.5 mM glycerol promoted the best growth of this microalga (1.19 × 10^6^ cells mL^−1^) (*p* < 0.05), even exceeding cell density obtained in photoautotrophic cultures (IM) (Figure 1). In addition, the higher concentrations of glycerol yielded the lowest cell growth. Table 1 shows the lower contents for Chl *a*, similar for Chl *b*, and higher for βCar in this alga under 12.5 mM glycerol compared to IM.

### 2.2. Combined Effect of Glycerol, Light, and Salinity on Mixotrophic Culture of D. salina

Figure 2a–d shows the effect of lower glycerol level combined with different light intensities and salinities on the growth and pigments of *D*. *salina*. The highest cellular yields were obtained with 12.5 mM glycerol, 3.0 M non-isosmotic salinity, and 50 μmol photons m^−2^ s^−1^ light intensity (Figure 2c) (*p* < 0.05), which exceeded threefold and twofold the microalgal productivity of the controls (IM) under 3.0 and 1.5 M NaCl, respectively (Figure 2a,c). The remarkable increase in chlorophylls *a* and *b*, and βCar (Table 2) was observed when *D*. *salina* was grown under the optimal glycerol treatment (12.5 mM glycerol, 3.0 M non-isosmotic salinity and 50 μmol photons m^−2^ s^−1^ light intensity). The analysis of culture pH in *D*. *salina* under the optimum glycerol treatment tended to mild basicity (7.0–8.2) (Table 2).

To stimulate the massive synthesis of β-carotene, *D. salina* cells that had reached their maximum at the seventh day of culture on inorganic medium enriched with glycerol were exposed to higher light intensity (from 50 to 300 µmol photons m^−2^ s^−1^) [30] for a period of three days. Figure 3 and Figure 4 show changes in the pigment content of *D. salina* under mixotrophic cultures at changed light intensity. In glycerol treatments under high light, chlorophyll *a* and *b* levels remained almost constant, whereas their levels of β-carotene and zeaxanthin increased compared to those treatments under low light.

Another set of experiments was developed to rule out whether the effect of glycerol by *D*. *salina* is dependent on change in salinity (1.5 to 3.0 M, non-isosmotic salinity) and not of the saline adaptation (3 M, isosmotic salinity). The microalga *D. salina* on 3 M isosmotic salinity exhibited intermediate cell growth. Chl *a*, Chl *b*, and BCar concentrations of microalgal cells exposed to this isosmotic salinity were lower than those obtained with non-isosmotic salinity, 3 M (Table 2 and Table 3). The culture pH under isosmotic salinity was maintained to be slightly acidic compared to the slightly basic pH of the microalgal culture under non-isosmotic salinity.

### 2.3. Extracellular and Intracellular Glycerol of D. salina 

Another group of experiments was carried out in order to know the dynamics of uptake of glycerol by *D*. *salina*. A continuous and substantial decrease in the extracellular glycerol of *D*. *salina* is observed with 3.0 M salinity compared to 1.5 M (Figure 5b). At the same time, a tiny amount of intracellular glycerol was observed throughout the microalgal culture at 1.5 and 3.0 M saline condition (Figure 5d). Lower values of extracellular (Figure 5a) and intracellular (Figure 5c) glycerol were observed in the inorganic medium without glycerol for both saline conditions. From these results, the uptake of glycerol from the medium by *D*. *salina* was preliminarily demonstrated.

### 2.4. PSII Activity from D. salina Cultivated on Glycerol

The maximum quantum efficiency Fv/Fm may be used to screen stressful or beneficial of organic substrates for the photosynthetic efficiency of *D*. *salina* under the mixotrophic growth conditions. In this work, the PSII activity of *D*. *salina* cultivated on glycerol is affected by the salinity level (Figure 6a,c). The Fv/Fm of *D*. *salina* cells cultivated on the optimized conditions of glycerol and light is greater for salinity 3 M than 1.5 M, but these are less than photoautotrophic cultures (Figure 6a,c). Microalgal cultures under glycerol and DCMU at 1.5 and 3 M NaCl showed a notable drop in Fv/Fm of *D*. *salina* cells for both salinities (Figure 6b,d). Furthermore, the different glycerol cultures and IM exposed to DCMU did not show microalgal growth.

## 3. Discussion

In this study, the combined effect of glycerol level, light intensity, and salinity was assessed in order to optimize the mixotrophic culture conditions of *D*. *salina*. Our main aim was to contribute to develop a simple mixotrophic strategy to increase biomass and β-carotene production of this halophilic chlorophyte based exclusively on the optimization of the mixotrophic culture conditions with a non-fermentative organic carbon substrate, *viz*., glycerol. The proper growth of *D. salina* under low doses of glycerol is indicative of the efficient diffusion, active transport, or both of this substrate through the cell membranes. At higher concentrations, the glycerol could have an inhibitory effect on the growth and productivity of *D*. *salina* as reported [20]. Even high levels of glycerol may change the chloroplast ultrastructure, which reduces the excitation energy in PSII and decreases the photosynthetic activity and chlorophyll content [31]. Decreased concentrations of *D. salina* Chl *a*, Chl *b*, and β-Car in glycerol compared to photoautotrophic cultures (Control) may indicate that heterotrophic and myxotrophic culture conditions are progressing (Figure 5), as reported in *Chlorella sorokiniana* UTEX 1230 [32] and 1602 [33], respectively. These variations may respond to transition imbalances between photoautotrophy and heterotrophy, which cannot be discarded according to the experimental mixotrophic growth conditions, as recently reported [34,35]. Moreover, some organic substrates can change the pH of the medium and other properties such as viscosity, osmotic pressure, and gas–liquid transfer coefficient [36]. The pH close to neutral obtained from the microalgal mixotrophy assayed, may be demonstrative of a balanced system between photosynthesis (CO_2_ consumed and O_2_ released) and respiration (O_2_ consumed and CO_2_ released) fated to the glycerol assimilation for this chlorophyte [34].

The best results of the mixotrophic growth of this alga were obtained from low doses of glycerol (12.5 mM) and non-isosmotic salinity (3 M NaCl), whose interaction stimulates the uptake of glycerol by the microalgal cells [37]. Concerning light intensity, better results were achieved with 50 μmol photons m^−2^ s^−1^, which demonstrated that glycerol has broad performance under dim light in microalgae [38], and happened when *D. salina* was grown in mixotrophic conditions. Regarding to salinity, this green alga can grow in a range from 0.05 M to 5.5 M NaCl [25], but show maximal growth performance at 1.5–3.0 M NaCl [29]. Compared to cultures at 1.5 M NaCl, *D. salina* grown at 3.0 M NaCl favors higher growth and synthesis of β-carotene in this alga on glycerol due to its antioxidant ability [29]. The high cell density obtained at usual promoting salinities for the growth of *D. salina* demonstrated that exogenous glycerol might play a nutritional role in this alga besides its well-known osmoregulatory role. This advantage is supported by its lower energy cost for cell metabolism compared to the assimilation of endogenous glycerol from photosynthesis or starch degradation [39]. The conversion of glycerol into glyceraldehyde 3-phosphate (G3P) allows for the binding to precursors associated with the carbon metabolic pathways of photosynthesis and respiration (phosphoenolpyruvate or pyruvate) that can affect cell growth and the synthesis of cellular metabolites [40,41]. In the optimized conditions of this work, chlorophyll *a* level (4.58 pg. cell^−1^) was lower than the level of 11 pg cell^−1^ obtained from *D. salina* cultures with glycerol [20]. However, initial β-carotene levels (4.43 pg. cel^−1^) in this study were higher than β-carotene (0.8 pg cell^−1^) reported in this same work. Unlike the current knowledge on the mixotrophy of *D. salina* with glycerol, which is dominated by the photoautotrophic component, the optimized conditions of this work could induce glycerol balanced mixotrophy with codominance of both pigments.

In comparison to the straightforward mixotrophic production of biomass and β-carotene with different organic carbon substrates (glucose, acetate, and glycerol), our optimized cultivation condition with glycerol enhances suitable and productive balance between growth and carotenoids in this halophilic microalga (Table 4). In most initiatives on mixotrophic cultures [12,13,20], only microalgal growth is enhanced, but β-carotene concentration remains low. Unlike our work, most of the reported microalgal cultures with higher growth or pigment levels [10,12,13,20] are due to additional environmental and nutritional conditions that different authors carry out (e.g., higher substrate concentration, inoculum size, and high light intensity, longer cultivation periods, addition of other nutrients, and agitation or aeration systems), which affect microalgal production costs. Our approach seeks to develop a simple, practical, and inexpensive mixotrophic system. As a C3 molecule, glycerol can integrate into photoautotrophic and photoheterotrophic metabolism, but an efficient metabolic fit for its assimilation slightly favors photosynthesis over respiration that could be undergoing in this alga. 

In our case, the photosynthetic reactions in *D. salina* were able to continue producing chlorophyllian pigments, energy, CO_2_ fixation, and even the incorporated carbon from glycerol could have been used to synthesize compounds not directly involved with cellular respiration (e. g., amino acids synthesis) [42]. Moreover, since the oxidative metabolism of the organic substrate is operating simultaneously in mixotrophy, a part of C-glycerol may switch to the tricarboxylic acid (TCA) cycle, where citrate intermediates are expressed at the gene level to induce respiration via the alternative mitochondrial pathway [43]. In principle, this mechanism may facilitate the massive synthesis of carotenoids [43] as the increase in β-carotene observed in *D. salina.* (Table 2). 

The slightly basic pH values with glycerol are quite different from the sharp values of acidity and basicity obtained with glucose and acetate, respectively (data no shown). The mixotrophic growth of D. salina with 12.5 mM glycerol, 3.0 M non-isosmotic salinity, and 50 μmol photons m^−2^ s^−1^ light intensity suggests a balance in the following conditions: CO_2_/O_2_ gases, photosynthesis and respiration, and pH [34] during organic and inorganic carbon metabolism. Lastly, the highest microalgal productivity obtained under a self-balanced culture pH could provide comparative advantages of mixotrophic cultures with glycerol compared to other organic substrates (e.g., glucose and acetate) for the massive and commercial production of this microalga. 

In the case of *D. tertiolecta* cultivated under non-isosmotic conditions (0.5 to 4 mM NaCl), unlike the low assimilation rates of 5 mM glycerol (0.78 pg glycerol cell^−1^ min^−1^ for 5 min) under isosmotic conditions (0.5 mM NaCl), this alga could assimilate the substrate at a rate of 3.14 pg glycerol cell^−1^ min^−1^ for 15 min [37]. This phenomenon could also be occurring in our cultures with *D*. *salina*, where the cells grown under non-isosmotic salinity are receiving slight salt stress. This fact may trigger an osmoregulation process synthesizing the enzyme machinery involved with the uptake and assimilation of exogenous glycerol, fated to energy and carbon for osmotic regulation, growth, and cell pigments. Glycerol assimilation is species dependent; hence, some microalgae exhibit high growth in inorganic media supplemented with glycerol, while other species remain unaffected [44]. Unlike chlorophyte, photoheterotroph *Dactylococcus dissociatus* MT1 can totally consume the initial concentration of 6 mM of glycerol within 24 h [31]. According to our results, chlorophyte photoautotroph *D. salina* slowly and progressively consumes glycerol throughout the culture, and possibly according to its nutritional and osmoregulatory needs. Likewise, although the plasmatic membrane of *D. salina* is several orders of magnitude less permeable to glycerol osmolyte, the presence of this compound in the controls of our experiments may suggest that the leakage of glycerol through these membranes may be much more widely distributed than just the release of glycerol by death and cell lysis [45]. 

The entire fate of incorporated glycerol by *D. salina* is still unknown. In our work, the gradual increase in extracellular glycerol content was observed through the IM medium (5a), and increase and decrease in the glycerol of *D. salina* in the IM + Gly medium. We hypothesized that these fluctuations may obey to the maintenance of glycerol homeostasis by the microalgal cells. Regarding to the large difference between glycerol uptake and glycerol assimilated by cells for their osmoregulation, much glycerol is metabolized (through diffusion, active transport, or both) and incorporated into the networks and pathways involved in the production of biomass, β-carotene, and carotenogenesis in *D*. *salina*. However, further experiments would be required to confirm these assertions.

The maximum quantum efficiency Fv/Fm is an extremely sensitive, non-invasive index to assess the wellness of the microalgae photosynthetic apparatus via the photochemical efficiency of the PSII in response to energy metabolism and the interaction between carbon and nutrient assimilation [32,46]. Fv/Fm values obtained with glycerol (0.435 on average) were lower than those obtained in photoautotrophic cultures (IM), because the incorporation of this substrate linked to salinity could be generating different degrees of stress to the photosystems of *D. salina*. Contrary to the isosmotic condition (1.5 M), the non-isosmotic salinity (3 M) induces to the highest Fv/Fm of *D*. *salina* cells grown on this substrate under mixotrophy. Indeed, an increase in photosynthetic activity coupled to the use of glycerol for respiration could be produced by these saline changes as reported for other stressors (e.g., high light) [47].

A remarkable decrease in the Fv/Fmax index was registered when *D. salina* was cultured with glycerol and DCMU, reflecting a disruption in the energetic status of the photosynthetic intermediaries across the damage in PSII reaction centers from this chlorophyte during the combined action of photoautotrophy and mixotrophy. In some way, the exerted action by photosynthesis and glycerol metabolism allow to allocate carbon toward growth and metabolites synthesis [48]. In this mixotrophic process, energy-rich molecules such as G3P and dihydroxyacetone phosphate (DHAP) are generated from glycerol. These C3 molecules transform into pyruvate, which, in turn, is decarboxylated to convert into acetyl-CoA [49]. In microalgal mixotrophy, biomass (energy from TCA cycle) and metabolites (e.g., pigments and fatty acid) synthesis pathways compete for this key thioester substrate [50]. In this study, optimized mixotrophy conditions by glycerol suggest that acetyl CoA or another intermediate is appropriately channeled for mixotrophic biosynthetic processes that stimulate growth and β-carotene in *D. salina*. This is the first study that reveals an unknown role of glycerol in addition to that involved in osmoregulation, mainly the simultaneous and improved production of cellular biomass and pigments of *D. salina*, especially β-carotene. This metabolic pattern occurs when the particular mixotrophic conditions exerted by glycerol are optimized, particularly through a suitable and selective combination of salinity and light. The photoautotrophic cultures, such as in the case of *Dunaliella*, are characterized *per se* by the low capacity of microalgal photosystems to capture light energy, as well as inorganic carbon limitation in extreme and hypersaline environments and media. In contrast, the mixotrophic culture of *D*. *salina* when regulated by sodium chloride and glycerol may become a new culture strategy to increase *D*. *salina* production yields. This microalga cultured on glycerol exhibited maxima yields, *viz*., yield biomass/substrate (Yx/s), and yield product (β-carotene)/substrate (Yp/s) (Table 4). The bioproduction processes and parameters of microalgae and particularly of *Dunaliella salina* under mixotrophic growth conditions, still require of specific optimization approaches. Our mixotrophic cultures with glycerol did not receive CO_2_, any other source of nutrients (only the inorganic culture medium), or continuous mechanical agitation. In mixotrophic cultures, microalgal growth is affected by CO_2_/O_2_ availability from agitation [51]. Despite this, our Yx/s and Yp/s (β-carotene) results are superior to yields obtained by some reports on mixotrophic cultures in this species. (Table 4).

Microalgal cultures with the highest cellular growth (3.0 M NaCl and Gly) progressively reach high levels of β-carotene, which occurred in cellular cultures after three days of being subjected to a high light level. In *D. salina* CCAP 19/30 cells, this effect was attributed to an increase in light intensity (50 to 200 µmol photons m^−2^ s^−1^), which caused the stabilization of the photosynthetic apparatus and of the chlorophylls [47]. 

The increase in carotenoids (β-carotene and zeaxanthin) is related to their protective action against oxidative stress such as hypersalinity and high light intensity [52]. The microalga *Chlamydomonas acidophila* grown on glycerol seems to enhance the β-carotene pathway compared to its photoautotrophic metabolism [53]. Under these conditions, the acetyl-CoA and NADPH production required to increase the β-carotene content should come from the energy excess of glycerol catabolism as reported in another microorganism [54]. In addition to the bioconversion of exogenous glycerol into key intermediates of general metabolism, it could have directly impacted the synthesis of carotenogenesis. The effect of high intensity light has been related with the transcriptional activation of carotenogenic genes in response to stimulation by phytohormones [55]. Our results, in accordance with recent reports on high growth rates and β-carotene content in *D. salina* MCCS-001, are associated with the effect of indole-3-acetic acid [56] that could be derived from exogenous glycerol [57], which may support our results in cellular performance.

In most cases of photoautotrophic and mixotrophic cultures of *D. salina*, cellular density can improve and the content of β-carotene remains low, and vice versa (Table 5). Our culture system is more suitable and efficient for the production of biomass and β-carotene in *D. salina* compared with the current literature. Sohrabi et al. [20] obtained biomass yields higher than ours; however, their cultivation conditions consisted of 2× concentrations for the glycerol and inocula compared to our experimental approach. 

Until quite recently, it was generally believed that *D. salina* was an obligate photoautotroph. This assigned nutrition mode limited the development of the great potential of this model alga in many applications that remained unexplored and unexploited. Our results indicated that the synergy of photosynthesis and glycerol uptake under mixotrophic culture conditions induced higher yields of biomass and β-carotene when compared to the photoautrophic and heterotrophic metabolism of *D. salina*. In this research, glycerol, besides its main role in osmoregulation, plays a strategic role as a metabolic substrate particularly valuable for the profitable production of biomass and carotenoids, mainly β-carotene, at low cost, and opens new avenues for future applications inserting *D. salina* in the cost-effective production of various bioproducts, bioprocesses, and services. This approach makes mixotrophy in this alga advantageous in many marine, saline, and industrial production processes that are carried out in the presence of high concentrations of NaCl. Particularly, the advantages of this mixotrophy initiative represent a particular interest in halophiles, such as *D. salina,* for biofuel production in coastal–marine wastewater treatment and bioremediation that, when coupled to the mixotrophic production of low-cost biomass and β-carotene, represent a profitable and affordable approaches leading to biorefineries. The production of high-value nutraceuticals, antioxidants, pro-vitamin *A* production, food and feed pigments, and wastewater treatment and bioremediation can greatly benefit from this.

## 4. Materials and Methods

### 4.1. Strain and Culture Conditions

The Mexican strain, *Dunaliella salina* BC02 used in this work, was obtained from the microalgae collection of the Department of Marine Biotechnology from CICESE (Ensenada, Mexico). Microalgal cells were grown in 250 mL Erlenmeyer flasks containing 100 mL of inorganic culture medium (IM) designed for *Dunaliella* [60]. The inoculated culture flasks were placed in an environmental chamber built *ex professo* both to maintain the temperature of the culture (22 ± 02 °C) [61] and to manage the different light intensities with 40 watts cool white fluorescent lamps, unless otherwise indicated. Photoautotrophic pre-cultures were grown in IM at 50 µmol photons m^−2^ s^−1^ light, 1.5 M salinity, pH in the 7.5–8.0 range, and without aeration. Once at exponential phase, inocula precultures (10%) were set up at 0.1 × 10^6^ cells ml^−1^ for later inoculate each mixotrophic culture for *D. salina* with glycerol (Gly) and the controls.

### 4.2. Mixotrophic Growth Conditions Set Up 

To improve the β-carotene and microalgal growth, the proposed inorganic medium was optimized to the most suitable conditions of glycerol, light intensity, and salinity for the mixotrophic culture. The proper concentrations of 1.5 and 3.0 M NaCl used in the experimental units served to assess changes in cell growth by salinity according to Farhat et al. [29]. The light intensities were set according to Sforza et al. [62] and were set up as follows: heterotrophy in light (10 µmol photons m^−2^ s^−1^), mixotrophy (50 µmol photons m^−2^ s^−1^), photoautotrophy (100 µmol photons m^−2^ s^−1^), and carotenogenic light (300 µmol photons m^−2^ s^−1^) were measured with a light meter (VWR, Catalog Number 21800-014). A Hanna meter (HI98130) recorded the influence of pH on the cell growth in glycerol. The microalgal cultures were gently hand-shaken (two times daily for 2 min) [14], and CO_2_ unfed. The mixotrophic growth of *D*. *salina* was compared only to the photoautotrophic control, because no cellular growth on glycerol was observed under heterotrophic conditions without light. For mixotrophic experiments, glycerol (Sigma Aldrich, Mexico) was added in the inorganic growth medium (IM) to give the following concentrations: 3.12, 6.25, 12.5, 25, 50, and 100 mM, respectively.

### 4.3. Cell Growth and Dry Cell Weight

The cellular density of *D*. *salina* was evaluated by direct counting for each experimental treatment and control by the Neubauer chamber and an Olympus compound microscope model BX60. With the obtained data, a growth curve was made, which exponential phase was selected, in order to calculate the specific growth rate (µ) as suggested [63] from Equation (1):(1)$$\mu =\frac{\mathrm{lnN}1-\mathrm{lnN}0}{T1-T0}$$
where N1 and N0 are the final and initial cell density (cells number per mL), respectively; T1 and T0 are the final and initial cultivation time (days).

The initial and final biomass were determined gravimetrically to calculate the dry cell weight as recommended [64]. Aliquots of 10 mL of microalgal culture at each experimental condition were filtered through preweighed, and precombusted (450 °C for two hours) glass-fiber filters (Whatman GF/F, 47 mm, pore size 0.7 µm) using a vacuum pump (at 35 to 55 mm Hg). Then, the filters were washed three times, with 20 mL 0.5 M ammonium formate to remove the remaining salt. Finally, filters were dried and weighted to obtain dry biomass at constant weight.

### 4.4. Pigment Content

#### 4.4.1. Spectrophotometric Determination

For pigment extraction, 2 mL of microalgal suspension were placed in microtubes and centrifuged at 5000 rpm for 10 min at 4 °C in an Eppendorf refrigerate centrifuge. Then, the culture medium was removed, and 1 mL pure acetone was added to dissolve the residual pellet [65]. The samples were vortexed and stored at 4 °C in the dark for at least 5 h. Afterward, the homogenates were filtered onto 0.2 µm filters (Whatman) and the extracts were immediately used for spectrophotometric measurements. The analysis of chlorophyll *a* (Chl *a*), chlorophyll *b* (Chl *b*), and β-carotene (BCar) was carried out in a 2 mL quartz cuvette by a Hach DR 5000 spectrophotometer. The scan spectra from 200 to 700 nm, with a resolution of 1.0 nm, were digitally recorded and processed. The Equations (2)–(4) describe the calculated pigment concentrations as previously reported [66]: Later, these values were standardized using the cell concentration (pg cell^−1^) of each treatment.
Chl *a* = 11.75*A*662 − 2.35*A*645 (in µg mL of extract^−1^)(2)
Chl *b* = 18.61*A*645 − 3.96*A*662 (in µg mL of extract^−1^)(3)
(4)$$\mathrm{BCar}=\frac{1000A470-2.27\mathrm{Chl}a-81.4\mathrm{Chl}b}{227}\;(\mathrm{in}\;µg\;\mathrm{mL}\;\mathrm{of}\;{\mathrm{extract}}^{-1})$$

#### 4.4.2. HPLC Analysis

Pigment profile analysis and the quantitative estimation of chlorophylls and carotenoids were carried out using an Agilent 1260 HPLC system equipped with a Zorbax XDB-C8 reversed phase column (4.6 × 150 mm, 3.5 μm pore size and 1 mL min^−1^ flow-rate [67,68]. Pigments were extracted from 1 mL of microalgal culture using 1 mL of 100% acetone. The samples were placed at −20 °C overnight in darkness [68]. Prior to the HPLC analysis, samples were centrifuged at 5000 rpm for 5 min at 4 °C. The supernatant was filtered with 0.22 μm filter into amber vials. The pigments extracted in acetone were directly injected to the HPLC column. The column temperature was set at 60 °C and the injection volume was 20 µL. Mobile phase: solvent A, methanol and 28 mM tetrabutyl ammonium acetate, 70:30 (*v:v*); solvent B, methanol. Solvent delivery gradient was in % B (min): 5%, 0 min; 5%, 5 min; 95%, 22 min; 95%, 27 min; 5%, 30 min as reported [67]. Pigments were eluted at a flow rate of 1 mL min^−1^ and detected at an absorbance range of 360–700 nm, specifically at 436 nm to separate properly chlorophylls and carotenoids. The identification of the pigments was based on the comparison of the retention times of samples with those of pure standard pigments obtained from DHI Laboratory Products (Hørsholm, Denmark) as previously described [69].

### 4.5. Extracellular and Intracellular Glycerol of D. salina

Extra and intracellular glycerol was assayed enzymatically with a commercial glycerol determination kit (SKU, 12812, BioSystems Reagents & Instruments, Barcelona, Spain). For sample preparation, 1 mL aliquots of microalgal culture were placed into microtubes to determine extracellular and total glycerol [70]. A set of tubes was directly centrifuged at 5000 rpm for 15 min at 4 °C and the supernatants used to estimate extracellular glycerol according to manufacturer´s specifications. Other tubes with microalgal culture were incubated for 10 min at 100 °C. After, the homogenates were centrifuged and supernatants were used for total glycerol determination. Intracellular glycerol was calculated from the difference between total glycerol and extracellular glycerol. Glycerol concentrations were normalized to dry cells weight.

### 4.6. Photosynthetic Efficiency Measurement

The Chl *a* fluorescence index, represented by the equation: Fv/Fm = (Fm − F)/Fm [46] was applied to assess the changes in photosynthetic activity, *viz*., the maximum quantum efficiency of PSII in microalgal cells when exposed to glycerol. The Fv/Fm was determined on cell cultures of *D*. *salina* exposed to the best treatment (12.5 mM Gly, 50 µmol photons m^−2^ s^−1^ and 3.0 M NaCl), DCMU (3-(3,4-dichlorophenyl)-1,1-dimethylurea), and controls, respectively. The registered fluorescence was carried out in samples previously adapted to dim light for 15 min, at 25 °C, using a Pulse-amplitude modulated fluorometer (DUAL-PAM 100, Heinz Walz, Effeltrich, Germany). Triplicate measurements for each sample and five saturating pulses at intervals of 30 s were sufficient to obtain a stationary level of maximum fluorescence (Fm) of *D*. *salina* cells. The fluorescence signal was normalized to chlorophyll *a* content of the microalgal cells as recommended [46]. To evaluate the changes on PSII activity by glycerol, intact cells of *D. salina* were exposed to 20 mM DCMU [71].

### 4.7. Yield Parameters

Representative parameters of microalgal yield, such as: biomass/substrate yield (Yx/s) (g g^−1^) and β-carotene/substrate yield (Yp/s) (mg g^−1^) were calculated from Equations (5) and (6) as proposed [72]. The biomass productivity was calculated at maximum microalgal growth (seventh day), as follows:(5)$$\mathrm{Yx}/s=\frac{\mathrm{Produced}\;\mathrm{biomass}}{\mathrm{Consumed}\;\mathrm{substrate}}$$
(6)$$\mathrm{Yp}/s=\frac{\mathrm{Produced}\;\left(\mathrm{BCar}\right)\;\mathrm{product}}{\mathrm{Consumed}\;\mathrm{substrate}}$$

### 4.8. Statistical Analysis

All experimental designs accomplished the criteria of randomness to ensure regular and reproducible results as suggested [73]. The data correspond to the mean ± standard deviation from three independent cultures. One-way analysis of variance (ANOVA) at *p*-value < 0.05 was calculated to detect significant differences in microalgal growth and pigments under different concentrations of glycerol using the STATISTICA 12.0 program. A Tukey’s test was applied to assess differences among treatments means. The combined effect among glycerol concentration, salinity, and light intensity on cell growth, pigments, and culture pH were pondered by a factorial ANOVA test.

## 5. Conclusions

The photosystems of *D. salina* can choose other options than obligate photoautotrophy, viz., facultative mixotrophy. This microalga possesses the physiometabolic and productive capacity to assimilate flexible and highly reduced organic substrates as glycerol under optimized culture conditions. This condition induced by glycerol, light, and salinity serve to maximize β-carotene accumulation without compromising the high biomass production through dichotomous role of glycerol in photosynthesis and respiration under mixotrophy. In particular, the uptake of glycerol by *D. salina* represents more energy and carbon to promote growth, carotenoid synthesis, and carotenogenesis. This optimized mixotrophic culture system could be expanded to transfer strains and substrates, and increase production yields in a low-cost and sustainable way. In this research, glycerol, besides its main role in osmoregulation, plays a strategic role as a metabolic substrate in this microalga, and is particularly valuable for the profitable and affordable production of biomass and carotenoids, mainly β-carotene. It also opens new avenues for future applications inserting *D. salina* in the cost-effective production of various bioproducts, bioprocesses, and services. The synergy of photosynthesis and glycerol metabolism makes mixotrophy in this alga advantageous in many marine, saline, and industrial production processes that are carried out in the presence of high concentrations of NaCl. These advantages are of particular interest in halophiles, such as *D. salina,* for biofuel production in coastal–marine wastewater treatment and bioremediation that, when coupled to the mixotrophic production of biomass and B-carotene, represent profitable and affordable approaches. Particularly, this research may greatly impact the production of high-value nutraceuticals, antioxidants, pro-vitamin *A* production, the concept of marine algal biofuels and biorefineries, and wastewater treatment and bioremediation. 

## Figures and Tables

**Figure 1 metabolites-11-00866-f001:**
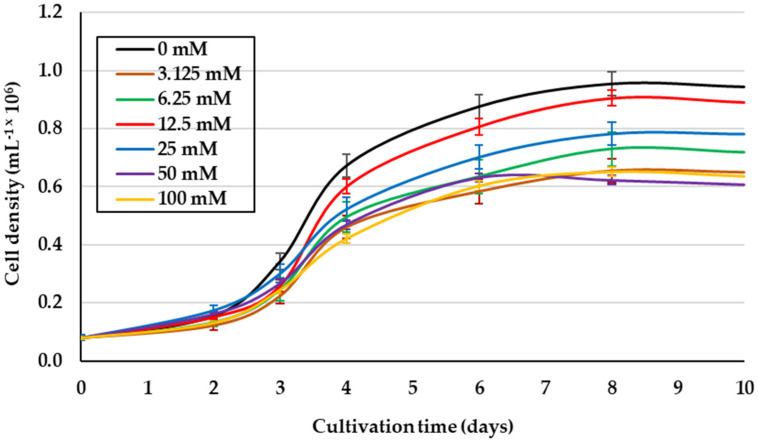
Cell density of *D*. *salina* cultured with different concentrations of glycerol (Gly). Salinity and light intensity were 1.5 M NaCl and 50 μmol photons m^−2^ s^−1^, respectively. Values correspond to the mean ± standard deviation of three replicates.

**Figure 2 metabolites-11-00866-f002:**
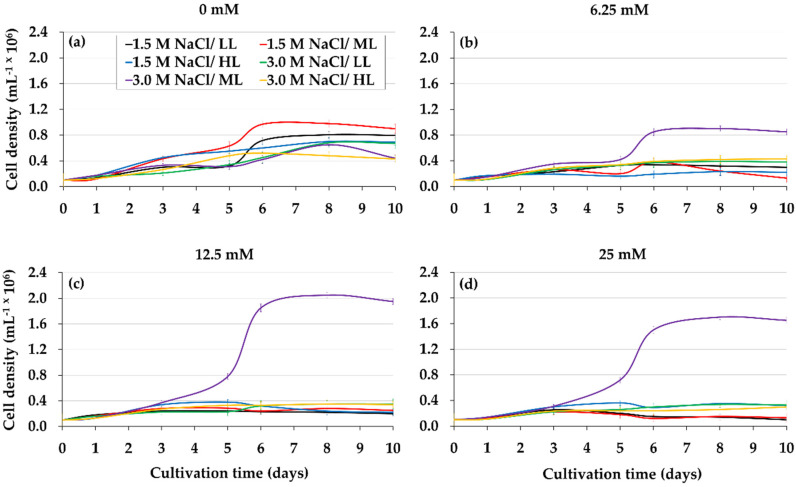
Cell density of *D*. *salina* cultured with different combinations of glycerol, salinity, and light intensity. The following interaction is indicated: glycerol, 0 mM (**a**), 6.25 mM (**b**), 12.5 mM (**c**) and 25 mM (**d**) with 1.5 and 3.0 M NaCl as well as with 10, 50 and 100 μmol photons m^−2^ s^−1^ respectively. Values correspond to the mean ± standard deviation of three replicates.

**Figure 3 metabolites-11-00866-f003:**
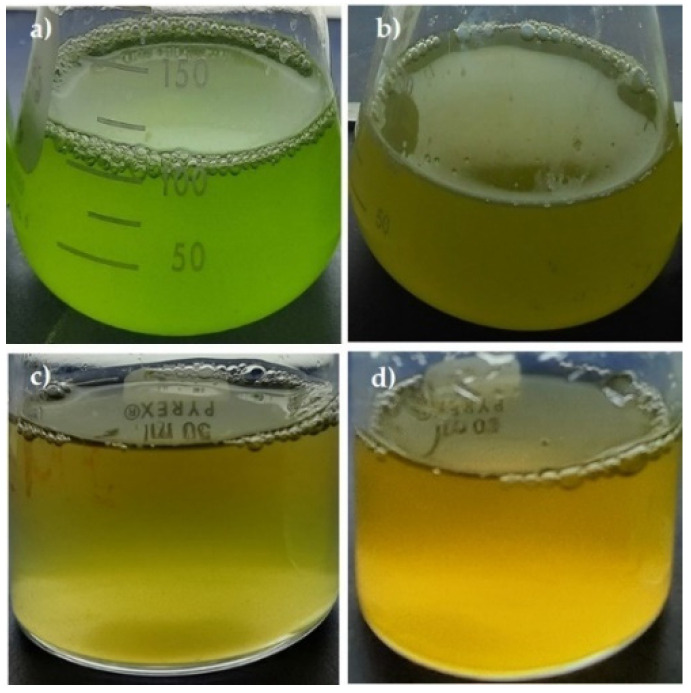
Cultures of *D. salina* on glycerol and controls. (**a**) IM, 1.5 M NaCl and 50 µmol m^−2^ s^−1^. (**b**) Gly, 3.0 M NaCl and 50 µmol photons m^−2^ s^−1^. (**c**) IM, 1.5 M NaCl and 300 µmol photons m^−2^ s^−1^. (**d**) Gly, 3.0 M NaCl and 300 µmol photons m^−2^ s^−1^. The images were captured on the seventh day of culture when the pigments were analyzed. (**a**) and (**b**) correspond to the original culture flasks, whereas (**c**) and (**d**) are recipients only used for photography.

**Figure 4 metabolites-11-00866-f004:**
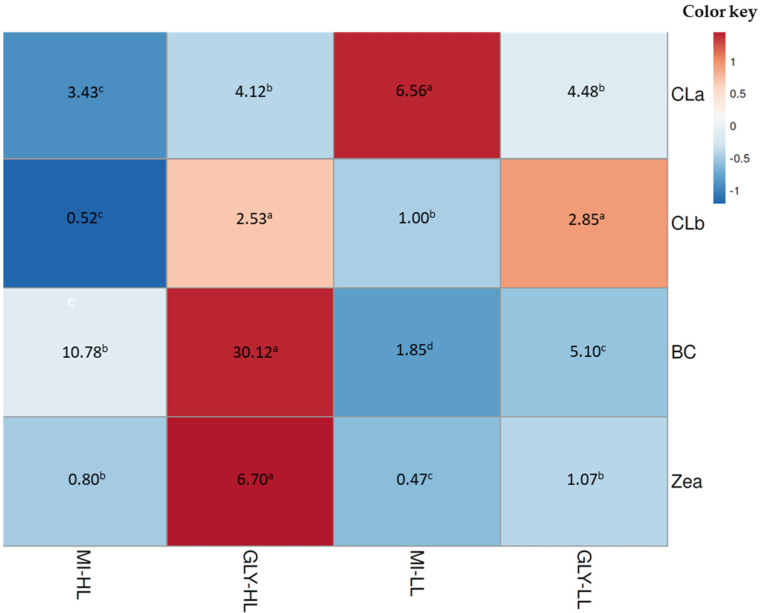
Heat map colors correspond to lower (blue) and higher (red) abundance for the main pigments of *D. salina* cultured under mixotrophic conditions. The real concentrations of pigments (pg cell^−1^) are in the heat map figure. MI-HL and Gly-HL correspond to the inorganic medium and glycerol exposed to 300 µmol photons m^−2^ s^−1^, respectively. MI-LL and Gly-LL correspond to the inorganic medium and glycerol exposed to 50 µmol photons m^−2^ s^−1^, respectively. Pigments (CLa: chlorophyll *a*, CLb: chlorophyll *b*, BC: β-carotene, Zea: zeaxanthin) were separated and identified by HPLC. Different letters in the same row are significantly different between pigment content for each treatment.

**Figure 5 metabolites-11-00866-f005:**
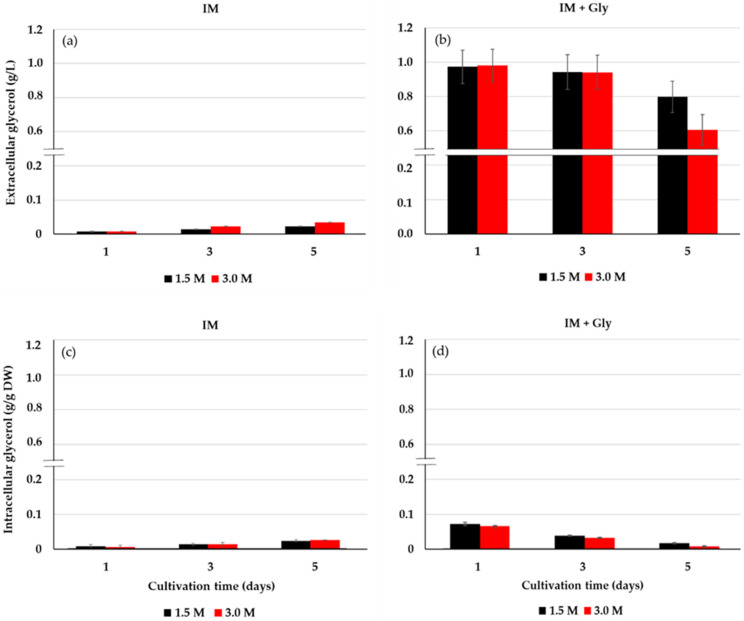
Extracellular glycerol from IM (**a**) and IM + Gly (**b**) and intracellular glycerol from IM (**c**) and IM + Gly (**d**) in *D*. *salina* cultivated on 12.5 mM glycerol. Salinities: 1.5 and 3.0 M NaCl. Intracellular glycerol concentrations were normalized to dry cell weight (DW). Values correspond to the mean ± standard deviation of three replicates.

**Figure 6 metabolites-11-00866-f006:**
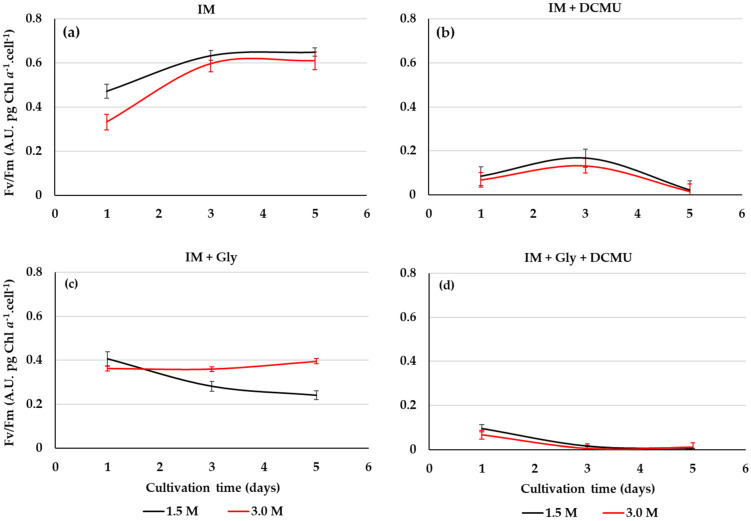
Photosynthetic efficiency (Fv/Fm) of *D*. *salina* cultivated at different conditions: IM (**a**), IM + DCMU (**b**), IM + 12.5 mM Gly (**c**), and IM + 12.5 mM Gly + DCMU (**d**). Culture salinities: 1.5 and 3.0 M NaCl. Values correspond to the mean ± standard deviation of three replicates.

**Table 1 metabolites-11-00866-t001:** Chl *a*, Chl *b*, BCar, and culture pH of *D*. *salina* grown under 12.5 mM Gly and photoautotrophic control. Efficient salinity for growth and light intensity were 1.5 M NaCl and 50 μmol photons m^−2^ s^−1^, respectively. Values in the same column with different superscript are significantly different.

Culture Medium	Chl *a* (pg cell^−1^)	Chl *b* (pg cell^−1^)	BCar (pg cell^−1^)	Final pH
IM	2.48 ± 0.18 ^a^	0.65 ± 0.05 ^a^	1.24 ± 0.09 ^b^	8.32 ± 0.11 ^a^
IM + 12.5 mM Gly	1.22 ± 0.09 ^b^	0.62 ± 0.05 ^a^	1.90 ± 0.14 ^a^	7.24 ± 0.05 ^b^
*p*-value	*p* < 0.00012	*p* > 0.495	*p* < 0.002	*p* < 0.00008

**Table 2 metabolites-11-00866-t002:** Chl *a*, Chl *b*, BCar, and culture pH of *D*. *salina* grown under different levels of glycerol, non-isosmotic salinity, and light intensity. The initial pH range was 6.20–6.35, and the final pH corresponds to the 7th day of cultivation. Values in the same columns with different superscript are significantly different.

Glycerol Level (mM)	Salinity (M NaCl)	Light Intensity (μmol Photons m^−2^ s^−1^)	Chl *a* (pg cell^−1^)	Chl *b* (pg cell^−1^)	Bcar (pg cell^−1^)	Final pH
0	1.5	10	3.95 ± 0.19 ^b^	1.21 ± 0.06 ^c^	1.56 ± 0.08 ^e^	8.55 ± 0.42 ^ab^
6.25	1.5	10	2.47 ± 0.12 ^c^	1.33 ± 0.07 ^c^	2.28 ± 0.11 ^d^	7.65 ± 0.33 ^b^
12.5	1.5	10	2.24 ± 0.11 ^d^	1.71 ± 0.08 ^b^	2.61 ± 0.13 ^c^	7.50 ± 0.37 ^b^
25.0	1.5	10	2.75 ± 0.13 ^c^	2.36 ± 0.11 ^a^	2.19 ± 0.11 ^d^	7.42 ± 0.38 ^b^
0	1.5	50	2.18 ± 0.11 ^d^	0.59 ± 0.02 ^d^	1.37 ± 0.07 ^e^	9.05 ± 0.45 ^a^
6.25	1.5	50	2.90 ± 0.14 ^c^	1.30 ± 0.06 ^c^	1.67 ± 0.08 ^b^	7.68 ± 0.38 ^b^
12.5	1.5	50	3.53 ± 017 ^b^	1.33 ± 0.06 ^c^	2.20 ± 0.11 ^d^	7.58 ± 0.37 ^b^
25.0	1.5	50	3.27 ± 0.16 ^b^	1.57 ± 0.07 ^b^	1.63 ± 0.08 ^e^	7.50 ± 0.35 ^b^
0	1.5	100	4.03 ± 0.20 ^ab^	1.05 ± 0.05 ^cd^	2.76 ± 0.14 ^c^	9.25 ± 0.46 ^a^
6.25	1.5	100	1.56 ± 0.08 ^e^	0.79 ± 0.04 ^d^	2.28 ± 0.11 ^d^	7.71 ± 0.38 ^b^
12.5	1.5	100	3.63 ± 0.18 ^b^	1.54 ± 0.07 ^b^	2.82 ± 0.14 ^c^	7.82 ± 0.39 ^b^
25.0	1.5	100	2.51 ± 0.12 ^c^	1.00 ± 0.05 ^cd^	2.66 ± 0.13 ^c^	7.63 ± 0.38 ^b^
0	3.0	10	2.47 ± 0.13 ^c^	1.64 ± 0.08 ^b^	2.16 ± 0.11 ^d^	8.38 ± 0.42 ^ab^
6.25	3.0	10	2.28 ± 0.11 ^d^	2.02 ± 0.10 ^a^	2.77 ± 0.14 ^c^	7.84 ± 0.39 ^b^
12.5	3.0	10	2.90 ± 0.15 ^c^	1.58 ± 0.08 ^b^	2.11± 0.10 ^d^	7.67 ± 0.38 ^b^
25.0	3.0	10	2.87 ± 0.14 ^c^	1.51 ± 0.08 ^b^	2.12 ± 0.11 ^d^	7.60 ± 0.38 ^b^
0	3.0	50	1.82 ± 0.09 ^e^	0.45 ± 0.02 ^d^	1.28 ± 0.06 ^f^	8.42 ± 0.42 ^ab^
6.25	3.0	50	2.48 ± 0.12 ^d^	0.75 ± 0.04 ^d^	2.22 ± 0.11 ^e^	7.28 ± 0.36 ^b^
12.5	3.0	50	4.58 ± 0.22 ^a^	1.40 ± 0.07 ^c^	4.43 ± 0.17 ^c^	8.10 ± 0.40 ^ab^
25.0	3.0	50	3.91 ± 0.19 ^b^	1.22 ± 0.06 ^c^	2.44 ± 0.12 ^d^	8.12 ± 0.42 ^ab^
0	3.0	100	2.77 ± 0.14 ^c^	0.63 ± 0.03 ^d^	4.54 ± 0.21 ^b^	8.82 ± 0.44 ^ab^
6.25	3.0	100	2.62 ± 0.13 ^c^	0.49 ± 0.02 ^c^	4.73 ± 0.23 ^ab^	8.84 ± 0.40 ^ab^
12.5	3.0	100	2.54 ± 0.12 ^c (1)^	0.44 ± 0.02 ^c^	4.51 ± 0.22 ^b^	8.78 ± 0.43 ^ab^
25.0	3.0	100	2.60 ± 0.13 ^c (2)^	0.57 ± 0.03 ^b^	5.03 ± 0.25 ^a^	8.72 ± 0.44 ^ab^
	*p*-value		*p* < 0.00002	*p* < 0.0001	*p* < 0.0002	*p* < 0.00002

**Table 3 metabolites-11-00866-t003:** Cell density, Chl *a*, Chl *b*, BCar, and culture pH of *D*. *salina* grown under 12.5 mM Gly and IM control. Isosmotic salinity for growth and light intensity were 3.0 M NaCl and 50 μmol photons m^−2^ s^−1^, respectively. The initial pH range was 6.20–6.35, and the final pH corresponds to the 7th day of cultivation. Values in the same columns with different superscript are significantly different.

Culture Medium	Cell Density (mL^−1^ × 10^6^)	Chl *a* (pg cell^−1^)	Chl *b* (pg cell^−1^)	BCar (pg cell^−1^)	Final pH
IM	0.82 ± 0.07 ^a^	1.02 ± 0.09 ^a^	0.15 ± 0.01 ^b^	2.39 ± 0.22 ^a^	7.85 ± 0.32 ^a^
IM + 12.5 mM Gly	0.98 ± 0.09 ^a^	1.05 ± 0.09 ^a^	0.57 ± 0.05 ^a^	2.24 ± 0.20 ^a^	6.52 ± 0.09 ^b^
*p*-value	*p* > 0.070	*p* > 0.712	*p* < 0.000	*p* > 0.426	*p* < 0.001

**Table 4 metabolites-11-00866-t004:** Comparison of yield parameters of *D*. *salina* grown in glucose and glycerol from the literature and in the best experimental conditions in this work.

Organic Carbon Substrate	Y_x/s_ (g g^−1^)	Y_p/s_ (mg g^−1^)	Reference
Glucose	--	8.91 ± 0.06	[11]
Glycerol	2.50 ± 0.08	1.16 ± 0.02	[20]
	2.33 ± 0.03	43.5 ± 0.08	In this work

**Table 5 metabolites-11-00866-t005:** Biomass and β-carotene of *D*. *salina* grown with various carbon substrates reported by the literature and in this work.

Substrate	Biomass	β-Carotene	Reference
Inorganic	1.0–5.0 × 10^6^ cells mL^−1^	4–8 pg cell^−1^	[58,59]
Glucose	0.99 g L^−1^	8.03 mg g^−1^	[12]
Acetate	1.0–1.2 × 10^6^ cells mL^−1^	<5 pg cell^−1^	[13]
Glycerol	1.0 × 10^6^ cells mL^−1^	0.94 pg cell^−1^	[28]
	4.0–5.0 × 10^6^ cells mL^−1^	0.8–1.0 pg cell^−1^	[20]
	2.15 × 10^6^ cells mL^−1^	30 pg cell^−1^	In this work
	(2.0 g L^−1^)	(50 mg g^−1^)	

## Data Availability

The data presented in this study are available in article.

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
