# Peer review of "Enhanced β-carotene and Biomass Production by Induced Mixotrophy in Dunaliella salina across a Combined Strategy of Glycerol, Salinity, and Light"

_metabolites, 2021, doi:10.3390/metabo11120866_

Round 1

Reviewer 1 Report

None

Author Response

Reviewer #1

Comments: none

Reviewer 2 Report

Review comments on “Special conditions of glycerol, salinity and light maximize biomass and β-carotene production of Dunaliella salina under mixotrophy”

Specific comment:

Figure4 Legend: Include this information “The real concentrations of pigments are in the heat map figure”

4.4.2. HPLC analysis:

Re-write this part “Pigment profile and a quantitative estimation of chlorophylls and carotenoids were analyzed using a Agilent 1260 HPLC system equipped with a Zorbax XDB-C8 reversed phase column (4.6 x 150 mm, 3.5 μm pore size and 1 ml min-1 flow-rate;” as “Pigment profile analysis and the quantitative estimation of chlorophylls and carotenoids were carried out using an Agilent 1260 HPLC system equipped with a Zorbax XDB-C8 reversed phase column (4.6 x 150 mm, 3.5 μm pore size and 1 ml min-1 flow-rate;”

Why do you need “(V/V)” expression while it was only 100% acetone?

Correct the spelling as “amber vials”.

Check if the injection volume was 1 µl instead of 1 ml?

Mention clearly if the pigments extracted in acetone was dried first? and then was re-dissolved in another solvent? Which means did you inject acetone containing pigments to the HPLC column or injection solvent was something else?

Lines 458, 469, and 538: Check the spelling “micromicromicroalgal”

Line471: Replace “water” with “culture medium”

Line473: Change “After the homogenates were filtered” to “Afterward, the homogenates were filtered”

Line478: Check for correctness here “(3) y (4) describe”

Author Response

Specific comment:

Figure4 Legend: Include this information “The real concentrations of pigments are in the heat map figure”

Answer: already corrected on the MS

4.4.2. HPLC analysis:

Re-write this part “Pigment profile and a quantitative estimation of chlorophylls and carotenoids were analyzed using a Agilent 1260 HPLC system equipped with a Zorbax XDB-C8 reversed phase column (4.6 x 150 mm, 3.5 μm pore size and 1 ml min-1 flow-rate;” as “Pigment profile analysis and the quantitative estimation of chlorophylls and carotenoids were carried out using an Agilent 1260 HPLC system equipped with a Zorbax XDB-C8 reversed phase column (4.6 x 150 mm, 3.5 μm pore size and 1 ml min-1 flow-rate;”

Answer: Corrected direct on the MS

Why do you need “(V/V)” expression while it was only 100% acetone?

Answer: already corrected on the MS

Correct the spelling as “amber vials”.

Answer: already corrected on the MS

Check if the injection volume was 1 µl instead of 1 ml?

Answer: already corrected on the MS

Mention clearly if the pigments extracted in acetone was dried first? and then was re-dissolved in another solvent? Which means did you inject acetone containing pigments to the HPLC column or injection solvent was something else?

Answer: already corrected on the MS

Lines 458, 469, and 538: Check the spelling “micromicromicroalgal”

Answer: already corrected on the MS

Line471: Replace “water” with “culture medium”

Answer: already corrected on the MS

Line473: Change “After the homogenates were filtered” to “Afterward, the homogenates were filtered”

Answer: already corrected on the MS

Line478: Check for correctness here “(3) y (4) describe”

Answer: already corrected on the MS

Reviewer 3 Report

Manuscript ID: metabolites-1423311

Title:  Special conditions of glycerol, salinity and light maximize biomass and β-carotene production of Dunaliella salina under mixotrophy

This manuscript has some issue with experimental conditions as well as this very simple study, which not contribute to filed.

Firstly, title of MS not scientific, need be revised.

Last section of abstract not clear “Line 23-27.

Fig. 3 “Figure 3. Cultures of D. salina on glycerol and controls. a) IM, 1.5 M NaCl and 50 µmol m-2 s -1 . b) 157 Gly, 3.0 M NaCl and 50 µmol photons m-2 s -1 . c) IM, 1.5 M NaCl and 300 µmol photons m-2 s -1 . d) 158 Gly, 3.0 M NaCl and 300 µmol photons m-2 s -1 . The images were captured on the seventh day of 159 culture when the pigments were analyzed.” Showing that authors used different type of cultivation vessel, and then how the results can be comparable. Bot type of cultivation vessel have different volume to surface ratio, which can differ the growth pattern.

Table 2 also not up to date there are many more reports also available in literature. “https://doi.org/10.1016/j.bcab.2020.101771”

Authors mixed the culture manually “hand-shaken daily” is replicable ???

Author Response

Reviewer #3

This manuscript has some issue with experimental conditions as well as this very simple study, which not contribute to filed.

Firstly, title of MS not scientific, need be revised.

Answer: The title has been changed to: Enhanced β- carotene and biomass production by induced mixotrophy of Dunaliella salina across a combined strategy of glycerol, salinity and light

Last section of abstract not clear “Line 23-27.

Answer: Already corrected on the MS

Fig. 3 “Figure 3. Cultures of D. salina on glycerol and controls. a) IM, 1.5 M NaCl and 50 µmol m-2 s -1 . b) 157 Gly, 3.0 M NaCl and 50 µmol photons m-2 s -1 . c) IM, 1.5 M NaCl and 300 µmol photons m-2 s -1 . d) 158 Gly, 3.0 M NaCl and 300 µmol photons m-2 s -1 .

The images were captured on the seventh day of 159 culture when the pigments were analyzed.” Showing that authors used different type of cultivation vessel, and then how the results can be comparable. Bot type of cultivation vessel have different volume to surface ratio, which can differ the growth pattern.

Answer: Already corrected on the MS

Table 2 also not up to date there are many more reports also available in literature. https://doi.org/10.1016/j.bcab.2020.101771

Answer: The information in Table 2 is from our original results and is not intended to include information from the literature. However, we have taken your recommendation and it is included in table 5.

Authors mixed the culture manually “hand-shaken daily” is replicable ???

Answer: Already corrected on the MS

Round 2

Reviewer 3 Report

Authors significantly revised the MS. 

This manuscript is a resubmission of an earlier submission. The following is a list of the peer review reports and author responses from that submission.

Round 1

Reviewer 1 Report

The MS has to be re-written prior and answer questions prior to detailed review to clarify a few basic  questions:

The key question: was the medium axenic, free of bacteria, fungi and other contaminating organisms or heterotrophs? Maybe the glycerol was metabolized and degraded to CO2?

Was the culture aerated, mixed?

What was the supply of Carbon? CO2 gas, bicarbonate, other?

How the culture pH was maintained constant?

Is thee any evidence of heterotrophic glycerol uptake by using C14 or C13 glycerol?

Was there any dark glycerol respiration?

having the answers to the above I will be able to continue the review.

Reviewer 2 Report

Review comments on “Special conditions of glycerol, salinity and light maximize biomass and β-carotene production of Dunaliella salina under mixotrophy”

General comment:

Be consistent in using light intensity unit as “µmol photons m-2 s-1” throughout the manuscript.

Dunaliella salina is one of the microalgae species, hence use “microalga” instead of “Algae”

In some paragraphs there are certain common English as well as typographical errors, which need to be fixed by reading carefully and thoroughly. For examples: Line 168: Inject volume was 1 ml (generally, injection volume); Line292: “glicerol”

Specific comments:

Line167: “The pigments were extracted using 2 mL acetone for 5 hours in darkness.” Provide detail method of pigments extraction followed by drying of pigments (if any) before re-dissolving in appropriate solvent that was injected for HPLC separation.

Figure 3: Why all four types of chromatograms are not shown? results of which are color coded below but in three and not four? Can you normalize the Y-axis labels of the chromatograms so that relative content of pigments are clearly comparable. Provide color code information for all four conditions and indicate what are a, b, and c mean in the color code image.

Figure3 legend: Mention what chromatograms are these.

Move figure 4 above as Figure 3. And mention if these images were captured on 7th day of cultivation when pigments were analysed.

Figure5: Generally this results are interesting as we know that Dunaliella salina is a glycerol producer, and that is why in IM medium (5a) on day 5th there was increase in extracellular glycerol content. On the contrary, on day 1 there was increase in intracellular glycerol content in “IM+Gly” medium, which reduced on 5th day. Does this means, a glycerol homeostasis is maintained? Elaborate the same in discussion section. Also, possibly discuss if any glycerol transmembrane transporter activities are involved. Interestingly, the carotenoid content was analysed on 7th day when internal glycerol content was already reduced to normal. Explain if the temporal extracellular glycerol concentration was responsible for this effect (if at all it was a possibility of turning ON specific enzyme activity of carotenogenesis pathway?)?

Reviewer 3 Report

Manuscript ID: metabolites-1344983

Title:  Special conditions of glycerol, salinity and light maximize biomass and β-carotene production of Dunaliella salina under mixotrophy

In this study authors fails to explain the novelty and why is this study is important and what are the research gap in this filed and what can be application use of this study.

Introduction is inadequate and extensive revision needed before publication. Please also cite few new reference citing the application of D. salina.

https://www.sciencedirect.com/science/article/pii/S0045653521010249

https://www.sciencedirect.com/science/article/pii/S2212982019309709

Material and method section is appropriate and discussed very well.

Line 306, 354: “D. salina” should be italic through out the MS.

Results need to be compared with literature reports and should be summarized in Table. Also, all figures need be in color and with high resolution.

Conclusion section need to be revised, please provide the brief conclusion with key results of this study and also highlight the novelty of this study.

Author Response

Please the attachment
